# Morally Distressing Experiences, Moral Injury, and Burnout in Florida Healthcare Providers during the COVID-19 Pandemic

**DOI:** 10.3390/ijerph182312319

**Published:** 2021-11-24

**Authors:** Lourdes P. Dale, Steven P. Cuffe, Nicola Sambuco, Andrea D. Guastello, Kalie G. Leon, Luciana V. Nunez, Amal Bhullar, Brandon R. Allen, Carol A. Mathews

**Affiliations:** 1Department of Psychiatry, College of Medicine-Jacksonville, University of Florida, Jacksonville, FL 32209, USA; Steven.Cuffe@jax.ufl.edu (S.P.C.); Amal.Bhullar@jax.ufl.edu (A.B.); 2Department of Clinical and Health Psychology, College of Public Health and Health Professions, University of Florida, Gainesville, FL 32610, USA; nsambuco@phhp.ufl.edu; 3Department of Psychiatry, College of Medicine, University of Florida, Gainesville, FL 32610, USA; aguastello@ufl.edu; 4Department of Psychology, University of North Florida, Jacksonville, FL 32209, USA; n01445131@unf.edu (K.G.L.); n01363717@unf.edu (L.V.N.); 5Department of Emergency Medicine, College of Medicine, University of Florida, Gainesville, FL 32610, USA; brandonrallen@ufl.edu; 6Department of Psychiatry, Center for OCD, Anxiety and Related Disorders, College of Medicine, University of Florida, Gainesville, FL 32610, USA; carolmathews@ufl.edu

**Keywords:** healthcare providers, moral distress, moral injury, burnout, depression, leadership support, longitudinal

## Abstract

Because healthcare providers may be experiencing moral injury (MI), we inquired about their healthcare morally distressing experiences (HMDEs), MI perpetrated by self (Self MI) or others (Others MI), and burnout during the COVID-19 pandemic. Participants were 265 healthcare providers in North Central Florida (81.9% female, M_age_ = 37.62) recruited via flyers and emailed brochures that completed online surveys monthly for four months. Logistic regression analyses investigated whether MI was associated with specific HMDEs, risk factors (demographic characteristics, prior mental/medical health adversity, COVID-19 protection concern, health worry, and work impact), protective factors (personal resilience and leadership support), and psychiatric symptomatology (depression, anxiety, and PTSD). Linear regression analyses explored how Self/Others MI, psychiatric symptomatology, and the risk/protective factors related to burnout. We found consistently high rates of MI and burnout, and that both Self and Others MI were associated with specific HMDEs, COVID-19 work impact, COVID-19 protection concern, and leadership support. Others MI was also related to prior adversity, nurse role, COVID-19 health worry, and COVID-19 diagnosis. Predictors of burnout included Self MI, depression symptoms, COVID-19 work impact, and leadership support. Hospital administrators/supervisors should recognize the importance of supporting the HCPs they supervise, particularly those at greatest risk of MI and burnout.

## 1. Introduction

Even prior to the events of the global health emergency caused by COVID-19, workplace burnout, which is recognized in the ICD-11 as a diagnosable condition [1] has been a problem of increasing concern among healthcare providers [2]. Since the start of the pandemic, health care providers (HCPs) have experienced increased psychological and physical stress due to the burden of caring for very ill patients, often in the context of fear of contracting COVID-19 and spreading the virus to friends and family [3,4].

Providers have experienced increased professional demands, often with limited equipment [5] and minimal training on how to safely perform appropriate healthcare in the context of increased personal risk. These systemic factors may lead HCPs to believe that their own mental and physical health is disregarded by employers, which in turn can result in feelings of distrust and a decline in their physical and/or mental health [6].

Given the challenges and stressors that COVID-19 has imposed on the healthcare system, a focus on psychiatric symptomatology and burnout has become more prevalent in the last two years. For example, the results from a study conducted in a large tertiary hospital in central Israel indicate that those who worked in a COVID-19 unit, in comparison to a non-COVID-19 unit, experienced significantly higher rates of anxiety about spreading the virus to loved ones, mental exhaustion, and post-traumatic stress symptoms [7]. A study of Italian healthcare professionals (*N* = 1153) found that more than 1 in 3 reported high levels of emotional exhaustion and 1 in 4 reported high levels of depersonalization [8]. Because these symptoms of burnout may negatively affect the quality of healthcare delivery [9,10], there is concern for the HCPs mental health. The increased mental health difficulties and burnout of the HCPs may relate to moral distress as has been reported in prior research not focused on medical professionals [11].

It is critical to differentiate the concepts of moral distress and moral injury (MI), which are related but separate entities. Moral distress is present when one is aware of the right thing to do but is unable to do so because of occupational constraints. In the context of the COVID-19 pandemic, HCPs may be experiencing what we term as healthcare morally distressing events (HMDE), such as being unable to provide frequent enough care, conduct necessary procedures or assessments, and refer patients for necessary procedures or specialists. Additional stressors may be present for providers doing telemedicine, such as discomfort with the use of telemedicine or belief that they are providing inadequate care. These events may lead to MI, a construct well studied within military populations [12,13] but less investigated among HCPs. MI occurs when one’s deeply held moral beliefs have been transgressed by perpetrating, failing to prevent, or witnessing what are considered by the individual to be immoral acts, thus producing a lasting psychological, biological, spiritual, behavioral, and social impact [12]. Individuals experiencing MI may feel shame and guilt, emotional distress, weakened trust, reduced self-forgiveness, view of self as immoral/irredeemable in an unjust world, and suicidality [12,13]. MI can have tragic consequences and has been implicated in the increased rates of suicide among HCPs, including the EM physician Lorna Breen, who struggled in making life-and-death decisions about which patients to treat and where to allocate the limited resources [14].

COVID-19 has drastically influenced the delivery and quality of care. The negative limitations in both delivery and quality of care are events considered as morally distressing. When compared to other morally distressing events, inadequate care was most reported and suggestive of moral injury in over 80% of 6026 frontline healthcare workers. The feelings of inadequacy of care and incompetency of their work abilities was also highly related to work difficulties, being another indicator of burnout [3]. Recognizing the specific HDMEs that contribute most to MI can be helpful at identifying appropriate coping factors or support systems.

A study completed prior to the pandemic focused on a sample of 181 Central North Carolina health professionals and found that 23.9% of providers were experiencing moderate functional impairment related to MI [15]. Bivariate analyses indicated that the significant predictors of MI were lack of religious affiliation, less religiosity, younger age, shorter time in practice, frequency of medical errors committed in the past month (i.e., mistakes that had the potential to cause harm to a patient, whether or not they actually did), and extent of depression, anxiety, and burnout symptoms. Psychological symptomatology was also shown to have a relationship with the development of MI.

As could be expected, there have been calls to investigate MI in healthcare workers during COVID-19 [16,17,18,19]. Two publications looked at the same sample of 838 healthcare workers from Maryland and found that MI remained stable over time, was associated with sleep disturbance, and was mitigated by a supportive workplace [20]. A more recent study conducted on 109 healthcare workers recruited online through personal contacts and professional listservs focused on predicting MI from subscales of a measure of professional quality of life [21]. They found that compassion satisfaction, burnout, and secondary traumatic stress were correlated with MI and that secondary traumatic stress was the only significant predictor when all were entered into multiple regression analysis.

There is a need to understand which risk factors may be associated with increased MI. Because moral distress relates to the inability to do what is believed to be appropriate for healthcare, it may be that health care providers with less decision-making power are most negatively impacted. Research prior to the pandemic suggests that nurses, because of their lower status and lack of authority, may feel helpless when critical decisions are made [22].

MI may also be impacted by predisposing risk factors, such as prior mental health and medical adversity, which may retune the autonomic nervous system and the brain in preparation for stress [23,24,25,26,27,28,29,30,31]. Prior research collected during the pandemic suggests that self-reported autonomic reactivity mediates the relationship between prior adversity and current worry, depression, and PTSD symptomatology [24]. Thus, HCPs with histories of prior adversity may be at greater risk, especially if they are lacking the personal resilience and leadership support to deal with the stress they are experiencing.

Thus, our study investigated the occurrence of HMDEs and the extent to which HCPs in a hospital system in North Central Florida were experiencing MI, both perpetrated by self (Self MI) or observed to be perpetrated by others (Others MI). We investigated whether MI was associated with specific HMDEs, such as the inability to see patients as frequently as felt to be necessary, conduct necessary assessments and procedures, and refer to specialists and for necessary tests. We also explored the potential contributions of risk factors (e.g., demographic characteristics, prior adversity, and COVID-19 protection concern, health worry, and work impact) and protective factors (i.e., personal resilience and leadership support) on both Self and Others MI. In addition, we explored whether the experience of MI was related to co-occurring psychiatric symptomatology (i.e., depression, anxiety, and PTSD). Lastly, we explored how Self/Others MI, psychiatric symptomatology, and the risk and protective factors related to burnout. We hypothesized the following:HCPs would report significant levels of Self/Others MI and burnout, and those experiencing more MI would report experiencing more burnout.HCPs reporting PMDEs and those impacted by prior adversities, less resilient, and feeling less supported by hospital leadership would be more likely to report MI.HCPs experiencing MI would be at greater risk of scoring above the clinical cutoff for psychiatric symptomatology.HCPs reporting more MI and psychiatric symptomatology, being more impacted by COVID-19 at work, and those experiencing less leadership support would be more likely to score above the clinical cutoff for burnout.

We also explored whether there would be differences in the moral injury and extent of burnout among providers in different healthcare roles (e.g., doctor, nurse, or assistant/technician).

## 2. Method

### 2.1. Participant Recruitment

The procedures used in this study were approved by the Institutional Review Board of the [edited out for blind review]. Potential participants were healthcare workers in two academic medical centers affiliated with a state university system in North Central Florida. One of the centers is a safety net hospital in a large city that receives some funding from the city to care for the indigent population, and the other center is a large tertiary care hospital in a mid-size city. During the data collection, there was a dramatic increase in rates of hospitalization at both sites, with the COVID-19 caseloads exceeding capacity in the large city.

Potential participants were invited to join this longitudinal study via flyers distributed in hospitals, nursing homes, and outpatient clinics in two cities in North Central Florida. Within the two academic hospitals in the region, healthcare workers were also emailed a brochure explaining the study from the department head or administrator. Participants were eligible for the study if they worked in a healthcare setting, regardless of their type of employment. Participants that consented to participate and completed measures via REDcap were compensated in an escalating manner for number of sessions completed. The total possible compensation was USD 70 for completion of all possible assessments over a four-month period and USD 220 for completion of all possible assessments over the total eight-month period. The analyses described in this study focus on those participants who identified as HCPs. Thus, hospital workers such as patient sitters, administrative support staff, and food service workers, while eligible for participation in the larger study, were not included in these analyses. Data for each participant were collected via Redcap at four time points from October 2020 to March 2021. The analytic team was blinded to the identity of the participants.

### 2.2. Constructs and Measures

**Healthcare morally distressing experiences.** Participants responded to four questions created for this study that related to their perceived inability to provide optimal care in the context of the COVID-19 pandemic. Specifically, they were asked whether they were able to provide care to patients at appropriate frequency, conduct necessary assessments or procedures, refer patients to specialists, and refer patients for necessary procedures. Two other questions completed by providers doing telemedicine asked about level of comfort with telemedicine and perception of the quality of care being provided with telemedicine. For these items, the providers who disagreed (e.g., felt that they were unable to provide appropriate care) were considered to have experienced the healthcare morally distressing experiences (HMDEs).

**Moral injury.** The Moral Injury Events Scale [32] assessed only the level of agreement via a 6-point Likert scale (0 = *strongly disagree* to 5 = *strongly agree*) about the occurrence/anguish of moral injury perpetrated by themselves and others, and not the perception of betrayal, which was excluded to limit the load on the participants. We focused on the internally consistent total score for Self MI (i.e., acting against moral or failing to act consistent with morals and feeling troubled by it; *α* = 0.94) and Others MI (i.e., seeing something morally wrong and feeling troubled by it; *α* = 0.88) when predicting burnout. For analyses focused on determining the factors related to moral injury we grouped the participants according to whether or not they agreed that they experienced Self and Others MI.

**Risk factors.** The risk factors considered included prior history of adversity and COVID-19 work factors. Prior adversity history was assessed via the Adverse and Traumatic Experiences Scale [33] which includes 30 items answered via a 5-point Likert scale (0 = *event did not occur*, 1 = *occurred and no impact on my life*, 2 = *minimal impact on my life*, 3 = *some impact on my life*, and 4 = *big impact on my life*). We focused on how impacted they were by their mental health adversity (*α* = 0.83) and medical health adversity (*α* = 0.52 because it only includes 3 questions related to different medical problems).

Regarding the COVID-19 work factors, the participants indicated the level of their COVID-19 health worry (*α* = 0.80) and COVID-19 work impact, which relates to how much COVID-19 had impacted their work life (such as whether they cared for patient who died from COVID-19; 6 items, *α* = 0.80) via a 5-point Likert scale (0 = *event did not occur* to 4 = *big impact on my life*). Participants also answered two questions regarding their access to personal protective equipment and supplies via a 4-point Likert scale (0 = *strongly disagree* and 3 = *strongly agree*) which formed the internally consistent (*α* = 0.80) COVID-19 protection concern. Participants also indicated whether they had been diagnosed with COVID-19.

**Protective factors.** We focused on two protective factors, personal resilience and leadership support. Personal resilience, which is the effective use coping strategies in flexible, committed ways to solve problems despite stressful circumstances, was assessed via the Brief Resilient Coping Scale [34] which is an internally consistent (*α* = 0.63) 4-item measure answered via a 5-point Likert Scale (1 = *does not describe me at all*, 2 = *does not describe me*, 3 = *neutral*, 4 = *describes me,* and 5 *= describes me very well*). Leadership support was assessed via the Leadership Behavior Description Questionnaire [35] which is an internally consistent (*α* = 0.75) 14-item measure that asks participants about their perception of hospital administration at making/communicating decisions and incorporating employee feedback via a 5-point Likert scale (0 = *never* and 4 = *always*).

**Current psychiatric symptomatology.** Depression symptomatology was assessed via the Patient Health Questionnaire-9 (α = 0.88) [36] and anxiety symptomatology via the Generalized Anxiety Disorders-7 (α = 0.92) and the suggested clinical cutoff of 10 or greater was used [37]. PTSD symptomatology was assessed via the 8-item PTSD Checklist List –5 (PCL-5; *α* = 0.92) that uses a clinical cutoff of 19 or above [38].

**Workplace burnout.** This was assessed via the Professional Fulfillment Index (PFI) a 16-item measure that we used to measure HCPs’ attitudes about their work [39]. The burnout scale (*α* = 0.92) is made up of a work exhaustion subscale (*α* = 0.90) that assesses sense of dread, physical/emotional exhaustion, and lack of enthusiasm, and the interpersonal disengagement subscale (*α* = 0.90) that assesses empathy and connection with others, particularly patients and colleagues. Each item is scored on a 5-point Likert scale (0 = *not at all true* to 4 = *completely true*). We focused on the mean scores and used recommended clinical cutoff of 1.33 to determine providers scoring above the clinical cutoff for burnout [39].

### 2.3. Data Analysis

Data were analyzed using IBM SPSS Statistics for Windows, Version 26.0. Armonk, NY: IBM Corp). In addition to the descriptive statistics, individual logistic regression analyses assessed whether participants reporting specific morally distressing experiences and risk/protective factors were more likely to report Self and Others MI. Additional logistic regression analyses explored the contributions of both Self and Others MI to increasing the odds of scoring above the clinical cutoff for each of psychiatric symptomatology (depression, anxiety, and PTSD) and burnout. Multilinear regression analyses explored the contributions of Self and Others MI total scores, psychiatric symptomatology scores, and risk and protective factors in predicting the two components of burnout, level of exhaustion and disengagement. Forward conditional logistic regression analyses (using *p* < 0.05 as the inclusion cutoff) determined which factors differentiated between providers scoring below and above the clinical cutoff for burnout.

Multilevel modeling analyses determined whether the relationship between the independent variable of MI and the dependent variable of burnout changed over time (*N* = 265, 211, 180, and 135 for Times 1, 2, 3, and 4, respectively) [40]. MLM, which is robust in its ability to account for missing data and has lower Type 1 error rates compared to ANOVA or ordinary least squares (OLS) regression [41] allowed for the examination of both between-subject and within-subject variances. Time (i.e., session number) was entered both as a fixed and random effect to account for both average rates of change and individual differences in rates of change. To examine the relationship between participant differences in the effect of MI on burnout, Self and Others MI were entered as overall averages and time-invariant covariates (TIC) at level-2 (i.e., fixed effects). To account for individual differences from month to month, all variables were also included as mean centered variables (each data point had the participant’s mean for that variable subtracted from the value to create a variable that represents session to session variability) at level-1 (i.e., random effects) and level-2 (i.e., fixed effects). In order to determine the role of each predictor, a series of models were fitted. First, to establish that there is sufficient variability in the dependent variable to explain with additional predictors, an unconditional means model (Model A) was used to test whether there was significant between-subject and within-subject variance in the dependent variable, burnout. Next, additional models examined the impact of time/month (Model B), MI perpetrated by others and self (Model C), and the interaction terms between the significant predictors (*p* < 0.01) from model C and time (Model D). Model fit was determined by evaluating significant reduction in the −2 log likelihood (−2LL) coefficients as well as reductions in the Bayesian information criterion (BIC). Pseudo *R*^2^ (eta square) was calculated for within-subject and between-subject variance explained by calculating the difference between the residual variance present in the null model and the current model and then dividing by the residual variance in the null model.

## 3. Results

### 3.1. Participants

The participants were 265 HCPs that varied in age from 20 to 72 years of age (*M* = 37.62, *SD* = 11.08). As reported in Table 1, they also varied in their race, highest level of education, and yearly income. Most participants were female and married or in a long-term relationship (63.4%), and many had children under 18 years of age living in their homes (42.3%).

### 3.2. Healthcare Moral Distress and Injury

At the baseline assessment, Healthcare providers reported morally distressing experiences related to their patient care, including being unable to provide care to patients with appropriate frequency (14.3%), conduct necessary assessments/procedures (10.9%), refer patients to specialists (11.7%), and refer patients for necessary procedures/tests (13.5%).

Table 2 documents the reports of Self and Others MI by the HCPs, both with regard to their agreement that a moral violation occurred and that they were troubled by it. There was also a strong positive association between the intensity of their belief that a moral violation occurred and how troubled they were by it, with participant perceiving that a MI occurred being more troubled by the violation, Self MI *X*^2^(1, *N* = 265) = 188.10, *p* < 0.001 and Others MI *X*^2^(1, *N* = 265) = 151.69, *p* < 0.001. Although not presented here, we also found that the results were similar when we focused on their responses to these components of Self and Others MI. Thus, we focused on the report of Self and Others MI and found that six providers reported experiencing only Self MI, 61 providers reported experiencing only Others MI, and 21 of reported experiencing both. Participants experiencing Self MI were more likely to have experienced Others MI, *X*^2^(1, *N* = 265) = 30.86, *p* < 0.001.

### 3.3. Odds of Experiencing Moral Injury

Table 3 displays the results of logistic regression analyses that explored whether HMDEs and risk/protective factors impacted the odds of experiencing Self and Others MI. The inability to provide frequent care to patients increased the odds of experiencing both Self and Other MI, whereas the inability conduct assessments and discomfort providing telemedicine increased the odds of experiencing Self MI, and the inability to refer for tests increased the odds of experiencing Others MI.

There was variability with regard to the potential risk factors, as three predictors increased the odds of experiencing Self MI and many predictors increased the odds of experiencing Others MI. Specifically, providers who were more impacted by caring for COVID-19 patients, concerned about their COVID-19 protection, and experienced less leadership support were more likely to experience Self MI. Although these predictors also increased the odds of Others MI, there were other important predictors. The odds were also increased for nurses and providers who were impacted by their prior mental and medical health adversities, had more COVID-19 health worries, and had been infected with COVID-19. Of note, level of personal resiliency and the demographic factors did not significantly impact the odds of experiencing Self or Others MI.

### 3.4. Moral Injury, Psychiatric Distress, and Burnout

Many providers scored above the clinical cutoff for depression (24.1%), anxiety (24.9%), PTSD (11.7%), and burnout (44.4%). Those scoring above the clinical cutoff were more likely to have experienced MI. Logistic regression analyses with both Self and Others MI entered as predictors of likelihood of scoring above the clinical cutoff for depression, anxiety, and PTSD indicated that the experience of both Self and Others MI increased the odds of scoring above the clinical cut off for depression (Self OR = 2.43, 95% CI = 1.01–5.86, *p* = 0.047 and Others OR = 2.96, 95% CI = 1.59–5.54, *p* < 0.001; correct classification 77.0%). However, only Self MI was associated with an increased likelihood of scoring above the clinical cut-off for anxiety (OR = 3.84, 95% CI 1.60–9.22, *p* = 0.003; correct classification 76.2%), PTSD (OR = 6.10, 95% CI = 2.20–16.90, *p* < 0.001; correct classification 88.2%), and burnout (OR = 4.36, 95% CI = 1.63–11.67, *p* = 0.003; correct classification 61.3%).

As indicated in Table 1, providers reported higher levels of exhaustion than disengagement. Although these two components of burnout were correlated, *r* = 0.65, *p* < 0.001, they only shared 42.6% of variance, and thus were examined separately. The results of the linear regression analyses focused on understanding the impact of Self and Others MI, psychiatric symptomatology, and risk/protective factors, while controlling for potential demographic differences, are presented in Table 4. The results indicated that the Self MI, depression symptoms, COVID-19 health worry, and leadership support were the significant predictors of level of exhaustion, while Self MI, COVID-19 work impact, and leadership support were the significant predictors of level of disengagement. Thus, demographic characteristics, prior adversity, and current personal resilience were not significant predictors.

Forward conditional logistic regression analyses predicting burnout indicated the significant model, *X*^2^ = 96.82, *p* < 0.001, included Self MI (OR 1.48, *p* = 0.015; 95% CI 1.08–2.03), depression symptoms (OR = 1.24, *p* < 001; 95% CI 1.16–1.33), COVID-19 work impact (OR 1.06, *p* = 0.044; 95% CI 1.00–1.13), and leadership support (OR 0.96, *p* = 0.007; 95% CI 0.94–0.99), and correctly classified 75.3% of providers above and below the clinical cutoff for burnout. The odds ratios above suggest that Self MI was the variable associated with the greatest likelihood of experiencing burnout.

### 3.5. Longitudinal Relationship between Moral Injury and Burnout

Table 5 presents the results of MLM analyses which examine the relationship between Self and Others MI scores and burnout scores. The unconditional means model (Model A) indicated there was enough variability in the burnout measurements to justify additional analyses. Model B was a significant improvement in fit (−2 LL) from the unconditional means model, *X*^2^(2) = 8.64, *p* < 0.05. It shows that on average there is a small but significant increase in burnout over time *b* = 0.42, *p* < 0.05, *d* = 0.39. Model C was also an improvement in fit from Model B, *X*^2^(8) = 96.59, *p* < 0.001. It indicates that (1) providers who averaged more Others MI *b* = 0.97, *p* < 0.001, *d* = 0.54, and Self MI *b* = 0.52, *p* < 0.001, *d* = 0.43, experienced higher burnout; (2) providers who had higher Self MI at any given time point also had higher burnout at that time point *b* = 0.21, *p* < 0.05, *d* = 0.48; and (3) Others MI did not matter from month to month in relation to burnout. Lastly, Model D was not an improvement in fit from Model C, *X*^2^(3) = 0.13, *p* = 0.99, indicates that the addition of time interaction did not explain any additional variance and Model C should be retained as the best fitting model.

## 4. Discussion

This study focused on the experience of morally distressing events and the factors that relate to Self and Others MI in healthcare providers and in determining how Self and Others MI impact professional burnout (including exhaustion and disengagement) during the height of the COVID-19 pandemic. Given the concern about the increased risk of burnout and other negative consequences for HCPs, working in the healthcare field, it was important to investigate potential risk factors, especially those that can be mitigated with the aim of reducing burnout. We also hoped that the protective factors of personal resilience and leadership support would positively impact levels of MI and burnout.

As hypothesized, we found that over the first four months of this study, which was conducted at the height of the COVID-19 pandemic, HCPs reported very high rates of burnout (44%), with burnout rates increasing over time. At baseline, HCPs (33.2%) reported high levels of MI, both directly experienced (Self) and observed (Others).

To our knowledge, this study is the first to examine the relationship between Self and Others MI among HCPs, which we found to be correlated but best viewed as separate constructs. Although we do not know why the majority of reported MI was Others MI rather than Self MI, we note that many of the providers who endorsed Self MI also endorsed Others MI. Based on this observation, we hypothesize that perhaps the majority of providers who believed they violated their own morals also believed that this was true of others. It may be easier for the HCPs to acknowledge Self MI if they believe others are also having such experiences. Alternatively, it may be that at least some of the HCPs who reported experiencing Self MI and Other MI were blaming themselves in a belief that they did not do enough to intervene when observing others do what they viewed as morally wrong.

The concerning rate of MI may relate to the high rates of morally distressing experiences related to patient care (e.g., inability to see patient frequently). In addition, both Self and Others MI were associated with specific HMDEs, COVID-19 work impact, COVID-19 protection concern, and leadership support. Others MI was also related to adversity history, COVID-19 health worry, prior COVID-19 diagnosis, and nurse role. The latter finding may be because nurses are spending more time with the patients and feeling less control or authority over their care. Although these associations were found, it is likely that MI may be more related to the actual events that occurred and other factors not investigated in this study, such as levels of stress or prior training.

As hypothesized and similar to previous findings [21], we found that MI was related to burnout. We also found that MI was associated with higher levels of psychiatric symptoms including depression, anxiety, and PTSD, a finding that is consistent with prior research focused on HCPs [3,15] and military samples [42]. In our sample, providers who reported Others MI were almost three times more likely to score above the clinical cutoff for depression, while those reporting Self MI were over two times more likely to score above the clinical cutoff for depression, over four times more likely to score above the clinical cutoff for anxiety, and over 6 times more likely to score above the clinical cutoff for PTSD. Psychiatric symptoms played a significant role in burnout, particularly with level of exhaustion.

Our findings suggest that firsthand involvement in moral violations (e.g., Self MI) may be more impactful on psychological wellbeing than secondhand observation of moral violations (e.g., Others MI). We found that it was Self MI (and not Others MI) that was strongly associated with burnout, and that additional factors, particularly leadership support, presence of depression, and COVID-related factors, were also important in predicting burnout. Together, these predictors correctly classified 70.7% of HCPs as scoring above or below the clinical cutoff for burnout, suggesting that they have substantial clinical relevance. These factors should be further addressed, both with additional research and with interventions to ameliorate their impact. Contrary to prior research [43], we also found that personal resiliency was not related to or predictive of burnout, which implies that it did not serve as a significant protective factor.

The finding that Self MI was the variable associated with the greatest likelihood of having burnout is unique but not surprising, given that research with Chinese HCPs [4] that MI was highly correlated with burnout. Our longitudinal results also indicated that Self MI (but not Others MI) was associated with higher rates of burnout at the four time points and that burnout scores significantly increased over this time. To our knowledge, this is the only study that examines how burnout changes over the course of the pandemic and how it relates to the types of MI. Because Self MI may strongly impact psychiatric symptomology and burnout over time, future research on MI in HCPs should examine both the rates of Self and Others MI, as in our study they emerged as separate constructs that had different effects on emotional health.

Self MI and perceived leadership support were significant predictors of both exhaustion and disengagement. While exhaustion was also impacted by their COVID-19 health worry and depression symptoms, disengagement was also influenced by how impacted they were by COVID-19 at work. This finding is particularly concerning because disengaged employees are more likely to withdraw, have lower job performance, and leave their jobs [44]. Thus, it could be argued that disengagement, even more than exhaustion, has the potential to negatively impact the quality of patient care provided during high-stress times such as a pandemic.

Finally, to our knowledge, the finding that perceived support by hospital or system leadership was inversely associated with burnout among HCPs is unique and consistent with the finding that nurse leaders can influence levels of engagement and improve patient outcomes [45]. Our finding requires further research to better understand the characteristics of leadership that are the most important contributors to reducing burnout. Strengthening the suggestion that leadership support mitigates a poor work environment [3], our findings suggest that in healthcare settings, leadership quality and style are critical factors that contribute to the wellbeing of employees. Hospital administrators and other supervisors in a healthcare setting should be aware of the importance of providing support to the HCPs they supervise, particularly those who may be at greatest risk of MI and burnout.

### Limitations and Future Lines of Research

Although the current study has many strengths, including its longitudinal design and focus on HCPs in the southeastern US, it also has some limitations. For example, the location limits the generalizability of the findings to other regions in the country that may have been impacted by the COVID-19 pandemic differently. Although the total sample was reasonably large, as is the case for many longitudinal studies, we saw a drop in the participation over time. This attrition may have had an impact on our longitudinal findings, as power dropped over time. It also negatively affected the collection of prior adversity information, as some of this data were collected at time 2 to minimize the burden at time 1.

Additionally, there is a possibility the HCPs experienced morally distressing experiences not included in the current study, such as shortages of ICU beds, triaging of patients to other facilities, and withholding care due to lack of resources. They may also have experienced distress due to lack of resources, watching patients suffer, or feeling as if they or their team were incompetent [3,22]. Future studies should investigate these potentially morally distressing experiences and their contributions to Self and Others MI.

Another limitation is the exclusion of the betrayal questions from the MI scale to limit the load on the participants. In retrospect, this data may have contributed to our understanding of leadership support. It is likely that less leadership support would have related to greater betrayal.

## 5. Conclusions

The level of COVID-19 and the stress on the healthcare system in the areas investigated in this study were lower at that time than major hotspot areas of the pandemic. Despite this, we found high rates of both MI and burnout among HCPs at all four time points, and that MI was associated not only with high rates of burnout but also with clinically significant levels of psychiatric symptomatology and COVID-19 factors. We also found that HCPs who were more worried about getting COVID-19 and more negatively impacted at work by COVID-19 were particularly vulnerable and at greater risk of burnout. Our findings suggest the need to provide HCPs with stronger leadership support, particularly in the context of crisis situations such as the COVID-19 pandemic.

## Figures and Tables

**Table 1 ijerph-18-12319-t001:** Demographic characteristics of healthcare providers sample.

Characteristics	*N*	%		*N*	%
**Gender**			**Yearly Income**		
Female	218	81.9	<USD 20,000	10	4.0
Male	48	18.1	USD 20,000–USD 40,000	26	9.9
			USD 40,001–USD 60,000	45	17.0
**Race**			USD 60,001–USD 80,000	46	17.4
White	206	77.7	USD 80,001–USD 100,000	33	12.3
Non-White	59	22.3	USD 100,001–USD 200,000	68	25.7
			>USD 200,000	37	13.8
**Location**					
Large city	162	61.1	**Psychiatric Treatment**		
Small city	92	34.7	Therapy	22	8.3
			Medication	27	10.2
**Education**			Both	58	21.9
High school Degree	14	5.4			
College Degree	138	52.1			
Graduate Degree	103	39.0			

**Table 2 ijerph-18-12319-t002:** Frequency of moral injury and burnout at time 1.

	*M*	*SD*
**Self Moral Injury ^a^**	1.66	1.05
1. I acted in ways that violated my own moral code or values. (Violation)	1.56	0.98
2. I am troubled by having acted in ways that violated my own morals or values. (Troubled)	1.70	1.23
3. I violated my own morals by failing to do something that I felt I should have done. (Violaton)	1.67	1.11
4. I am troubled because I violated my morals by failing to do something I felts I should have done. (Troubled)	1.71	1.21
**Others Moral Injury ^a^**	2.54	1.45
1. I saw things that were morally wrong. (Violation)	2.55	1.53
2. I am troubled by having witnessed others’ immoral acts. (Troubled)	2.53	1.55
**Professional Exhaustion ^b^**	1.86	1.07
1. A sense of dread when I think about work I have to do	1.71	1.21
2. Physically exhausted at work	2.06	1.24
3. Lacking in enthusiasm	1.63	1.17
4. Emotionally exhausted at work	2.04	1.30
**Professional Disengagement ^b^**	0.97	0.82
1. Less empathetic with my patients	0.84	0.94
2. Less empathetic with my colleagues	1.09	1.08
3. Less sensitive to others’ feelings/emotions	1.01	0.96
4. Less interested in talking with my patients	0.86	1.00
5. Less connected with my patients	0.91	1.02
6. Less connected with my colleagues	1.14	1.06

*N* = 265; ^a^ Mean scores were calculated to facilitate comparisons and interpretation according to the 6-point scale (1 = *strongly disagree* and 6 = *strongly agree*). ^b^ 5-point scale (0 = *not at all*, 1 = *very little*, 2 = *moderately*, 3 = *a lot*, and 4 = *extremely*).

**Table 3 ijerph-18-12319-t003:** Results of binary logistic regression odds of experiencing self and others moral injury.

Factors	Self MI			Others MI		
	OR	95% CI	*p*	OR	95% CI	*p*
**Morally Distressing Experiences**						
Inability to Provide Frequent Care	2.92	1.18–7.26	0.021	2.60	1.29–5.24	0.007
Inability to Conduct Assessments	2.78	1.01–7.62	0.047	1.71	0.78–3.78	0.182
Inability to Refer to Specialists	1.83	0.64–5.24	0.262	1.97	0.92–4.23	0.080
Inability to Refer for Tests	1.61	0.57–4.60	0.372	2.36	1.15–4.87	0.020
Discomfort Providing Telemedicine ^a^	13.33	1.58–112.43	0.017	0.73	0.08–7.06	0.789
Perception of Inferior Care with Telemedine ^a^	2.00	0.20–20.51	0.559	1.00	0.18–5.49	10.00
**Demographic Characteristics**						
Age	0.97	0.93–1.01	0.100	1.00	0.98–1.03	0.732
Gender	1.72	0.68–4.35	0.249	0.82	0.41–1.65	0.579
Educational Level	0.81	0.44–1.50	0.501	0.85	0.57–1.28	0.434
Income	0.91	0.72–1.14	0.394	1.02	0.88–1.19	0.768
**Healthcare Role**						
Doctoral Level	0.56	0.20–1.57	0.268	0.66	0.36–1.22	0.184
Nurse	1.28	0.51–3.20	0.598	2.07	1.15–3.70	0.015
Medical Assistant/Technician	1.64	0.63–4.26	0.307	1.24	0.64–2.36	0.523
**Other Risk Factors**						
Impact of MH Adversity ^b^	1.03	1.00–1.06	0.062	1.03	1.01–1.05	0.012
Impact of Medical Adversity ^b^	1.04	0.83–1.31	0.734	1.18	1.02–1.37	0.025
COVID-19 Work Impact	1.08	1.01–1.16	0.033	1.08	1.03–1.14	0.002
COVID-19 Health Worry	1.13	0.69–1.85	0.623	1.41	1.03–1.94	0.034
COVID-19 Protection Concern	4.38	1.87–10.29	<0.001	3.53	1.80–6.91	<0.001
COVID-19 Diagnosis	2.74	0.93–8.10	0.067	2.99	1.28–7.00	0.012
**Protective Factors**						
Personal Resilience	1.01	0.84–1.20	0.950	1.09	0.97–1.23	0.153
Leadership Support	0.97	0.93–1.00	0.048	0.95	0.93–0.97	<0.001

*N* = 265; ^a^
*N* = 68 because less providers were providing telemedicine; ^b^
*N* = 204 because data was collected at time 2.

**Table 4 ijerph-18-12319-t004:** Results of multilinear regression analyses predicting level of exhaustion and disengagement.

Factors	Exhaustion			Disengagement		
	*B*	*T*	*P*	*B*	*T*	*p*
Self MI	0.19	2.26	0.026	0.20	2.16	0.033
Others MI	−0.13	−1.42	0.159	−0.09	−0.89	0.378
Depression Symptoms	0.45	3.63	<0.001	0.20	1.42	0.158
Anxiety Symptoms	0.02	0.20	0.846	0.07	0.53	0.600
PTSD Symptoms	0.05	0.38	0.705	0.20	1.48	0.141
**Risk Factors**						
Age	−0.03	−0.38	0.705	−0.02	−0.22	0.827
Gender	−0.01	−0.13	0.896	−0.05	−0.59	0.557
Educational Level	−0.12	−1.30	0.195	−0.09	−0.80	0.380
Income	0.07	0.81	0.418	0.06	0.60	0.538
**Healthcare Role**						
Doctoral Level	0.10	0.79	0.430	0.08	0.61	0.543
Nurse	0.11	0.94	0.350	−0.04	−0.31	0.756
Medical Assistant/Technician	0.03	0.25	0.802	−0.02	−0.17	0.863
**Other Risk Factors**						
Impact of MH Adversity ^b^	−0.13	−1.50	0.136	−0.02	−0.20	0.839
Impact of Medical Adversity ^b^	0.03	0.38	0.706	−0.03	−0.28	0.783
COVID-19 Work Impact	0.01	0.13	0.896	0.26	2.69	0.008
COVID-19 Health Worry	0.26	3.11	0.002	0.01	0.08	0.936
COVID-19 Protection Concern	0.08	0.98	0.328	0.02	0.24	0.810
**Protective Factors**						
Personal Resilience	0.03	0.39	0.695	0.03	0.41	0.681
Leadership Support	−0.27	−3.60	<0.001	−0.22	−2.72	0.008

^b^*N* = 204 because data was collected at time 2. *Exhaustion F*(20, 104) = 6.53, *p* < 0.001, *R*^2^ = 0.56 and *Disengagement F*(20, 103) = 4.55, *p* < 0.001, *R*^2^ = 0.47.

**Table 5 ijerph-18-12319-t005:** Results of multi-level modeling predicting burnout from moral injury.

	Model A (Unconditional Means Model)	Model B	Model C	Model D
**Fixed Effects**				
Time		2.61 **	2.70 **	1.57
Mean Self MI			3.56 ***	3.45 ***
Mean Other’s MI			4.50 ***	4.19 **
Centered Self MI			2.31 *	2.27 *
Centered Other’s MI			1.92	1.94
Mean Self MI * Time				0.36
Mean Other’s MI * Time				−0.19
**Random Effects**				
Time		1.0	1.03	1.04
Centered Self MI			1.49	1.51
Centered Other’s MI			0.32	0.35
**Fit Statistics**				
−2LL	5072.76	5064.12 *	4976.17 ***	4976.04
BIC	5092.72	5097.38	5049.35	5062.53
*R*^2^ Within		0.06 *	0.15 **	0.15
*R*^2^ Between		0.01 *	0.33 **	0.33

Significant improvement from previous model fit at * *p* < 0.05. ** *p* < 0.01. *** *p* < 0.001.

## Data Availability

Data are available on a collaborative basis upon request to the corresponding author. Data are not publicly available due to privacy restrictions.

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
