# Peer review of "Morally Distressing Experiences, Moral Injury, and Burnout in Florida Healthcare Providers during the COVID-19 Pandemic"

_ijerph, 2021, doi:10.3390/ijerph182312319_

Round 1

Reviewer 1 Report

The manuscript addresses a very important topic and is a timely contribution to the literature on the human face of the healthcare infrastructure in the midst of the ongoing pandemic.

In general, the article is well written and the research is well designed. However, it would be helpful for the title or abstract to include the geographical region of the research, and for the introductory section to also make reference to the geographical regions that prior research has covered (only Italy is mentioned as where one study was conducted). In addition, it would be useful for the authors to give a brief description of the healthcare setting in which the respondents worked--the kinds of infrastructure in this part of Florida, the degree of public/private finances for the healthcare system in this area, as well as the general pandemic situation in the area (average case loads or daily rates of infections/hospitalizations, etc.), in order to enable researchers in other areas to better visualize the conditions. (It would also be helpful to contextualize or at least add an explanatory sentence about the Lorna Breen case, instead of just mentioning her name.)

There are some small typos that need to be fixed (e.g., Page 5 Line 253, "diving" should probably be "dividing"; missing parentheses and typos in Table 2, etc.), and some grammatical issues (e.g., Page 6, Line 257-256: this sentence seems oddly constructed).

Author Response

REVIEWER 1

The manuscript addresses a very important topic and is a timely contribution to the literature on the human face of the healthcare infrastructure in the midst of the ongoing pandemic. In general, the article is well written and the research is well designed.

  • Thank you

However, it would be helpful for the title or abstract to include the geographical region of the research, and for the introductory section to also make reference to the geographical regions that prior research has covered (only Italy is mentioned as where one study was conducted).

  • Geographic region of our study (North Central Florida) was added to the title and mentioned in the abstract and introduction
  • We have added geographic regions when necessary in the introduction.

In addition, it would be useful for the authors to give a brief description of the healthcare setting in which the respondents worked--the kinds of infrastructure in this part of Florida, the degree of public/private finances for the healthcare system in this area, as well as the general pandemic situation in the area (average caseloads or daily rates of infections/hospitalizations, etc.), in order to enable researchers in other areas to better visualize the conditions.

  • This was done in the participant recruitment section.

(It would also be helpful to contextualize or at least add an explanatory sentence about the Lorna Breen case, instead of just mentioning her name.)

  • This was done.

There are some small typos that need to be fixed (e.g., Page 5 Line 253, "diving" should probably be "dividing"; missing parentheses and typos in Table 2, etc.), and some grammatical issues (e.g., Page 6, Line 257-256: this sentence seems oddly constructed).

  • These corrections were made.

Reviewer 2 Report

Dear sir, thank you to select me to review manuscript: Morally Distressing Experiences, Moral Injury, and Burnout in  Healthcare Providers during the COVID-19 Pandemic written by Dalle LP et al.  265 participants have been included into final analysis, most of them were women. The authors concluded that high rates of both MI and burnout among participants and  MI was associated not only with high rates of burnout, but also with clinically significant levels of psychiatric symptomatology.

I recommend only one minor change:  Table 1., please change < $200, 000 for < $20, 000

Author Response

REVIEWER 2

Dear sir, thank you to select me to review manuscript: Morally Distressing Experiences, Moral Injury, and Burnout in Healthcare Providers during the COVID-19 Pandemic written by Dalle LP et al.  265 participants have been included into final analysis, most of them were women. The authors concluded that high rates of both MI and burnout among participants and MI was associated not only with high rates of burnout, but also with clinically significant levels of psychiatric symptomatology.

  • Thank you

I recommend only one minor change:  Table 1, please change < $200, 000 for < $20, 000

  • This correction was made.

Reviewer 3 Report

Thank you for the opportunity to review the manuscript, Morally Distressing Experiences, Moral Injury, and Burnout in 2 Healthcare Providers during the COVID-19 Pandemic , submitted to the Int. J. Environ. Res. Public Health.  I found it very interesting. The methods are methodologically sound. I think the topic is important and interesting to identify the impact COVID-19 had in Healthcare Providers .The different sections of the study seem well addressed, although I offer some suggestions below.

- In Abstract section, the conclusion needs more clear,  please make one sentence with more emphasis on the results found in the study. Conclusion should be more strong. It would be important to create a hook for the reader so that it is clear why read more than the abstract.

- In the results section the number of tables seems excessive, however the information on their are interesting for the study.

- Table 1 needs be corrected in Yearly Income data.

- The conclusion section needs more emphasis. Please make more emphasis on the results found in the study. Conclusion should be more strong.

Author Response

REVIEWER 3

Thank you for the opportunity to review the manuscript, Morally Distressing Experiences, Moral Injury, and Burnout in 2 Healthcare Providers during the COVID-19 Pandemic, submitted to the Int. J. Environ. Res. Public Health.  I found it very interesting. The methods are methodologically sound. I think the topic is important and interesting to identify the impact COVID-19 had in Healthcare Providers .The different sections of the study seem well addressed, although I offer some suggestions below.

  • Thank you

- In Abstract section, the conclusion needs more clear, please make one sentence with more emphasis on the results found in the study. Conclusion should be more strong. It would be important to create a hook for the reader so that it is clear why read more than the abstract.

  • This was done. We have a more specific conclusion

- In the results section the number of tables seems excessive, however the information on their are interesting for the study.

  • All table are essential in order to simplify the explanation of the results.

- Table 1 needs be corrected in Yearly Income data.

  • This correction has been made.

- The conclusion section needs more emphasis. Please make more emphasis on the results found in the study. Conclusion should be more strong.

  • This was done.

Reviewer 4 Report

First of all, I would like to thank you for the opportunity to review this manuscript. It is a very interesting study that, as its title indicates, explores "Morally Distressing Experiences, Moral Injury, and Burnout in Healthcare Providers". Below are a number of comments and suggestions in case they might be helpful:

TITLE

Adequate. It expresses very well the main ideas to be developed.

ABSTRACT

It has a good internal structure. It is recommended to put a little more sociodemographic information such as the total number of participants, the average age (and standard deviation) and the percentage of women or men over the total. 

KEYWORDS

A number of appropriate and very useful keywords have been selected for the search of the article. As a suggestion, without this being obligatory, it is recommended to put the words in alphabetical order (unless the authors have decided another order, such as relevance to the study, etc). It is also recommended that at least one of the keywords refer to the research design.

INTRODUCTION

The first section could be called "Introduction" as is usual in many Journals. With respect to the content, the authors have made an adequate description of the subject matter to explain and justify the study, based on a closely related bibliography.

METHOD

The process is very well described as well as the instruments used. With respect to the latter, statistical information on their internal consistency and description has been included. The data analysis is also well developed and provides useful information for replicating the study.

RESULTS

The results presented by the authors are closely related to the line of work proposed and its objectives. The section on Participants (together with Table 1) could be included within the Method since it performs a descriptive analysis of the sample. The Tables are clear and allow understanding of the results.

DISCUSSION

The authors discuss the results, assessing the verification of the hypotheses and comparing the information with other studies in an appropriate manner. Since a multi-author approach has been used in the theoretical framework (introduction), more use could be made of this valuable resource in the discussion as well, so it is encouraged to use more references than those previously cited in the Introduction in the Discussion if possible.

The limitations are very well developed and allow us to analyze the work and the subject matter from a broader and more mature point of view. This subsection -just as a comment- could be called "Limitations and future lines of research".

CONCLUSIONS

The inclusion of this section is considered very opportune. 

REFERENCES

Adequate. They are in line with the subject matter.

In summary, this is a very interesting work, with a good theoretical and empirical foundation. 

Author Response

REVIEWER 4

First of all, I would like to thank you for the opportunity to review this manuscript. It is a very interesting study that, as its title indicates, explores "Morally Distressing Experiences, Moral Injury, and Burnout in Healthcare Providers".

  • Thank you

Below are a number of comments and suggestions in case they might be helpful:

TITLE

Adequate. It expresses very well the main ideas to be developed.

  • Thank you

ABSTRACT

It has a good internal structure. It is recommended to put a little more sociodemographic information such as the total number of participants, the average age (and standard deviation) and the percentage of women or men over the total.

  • This was done

KEYWORDS

A number of appropriate and very useful keywords have been selected for the search of the article. As a suggestion, without this being obligatory, it is recommended to put the words in alphabetical order (unless the authors have decided another order, such as relevance to the study, etc). It is also recommended that at least one of the keywords refer to the research design.

  • We added longitudinal to the keywords. We kept the order of our key words to be from major focus of paper to lesser focus.

INTRODUCTION

The first section could be called "Introduction" as is usual in many Journals. With respect to the content, the authors have made an adequate description of the subject matter to explain and justify the study, based on a closely related bibliography.

  • Thank you

METHOD

The process is very well described as well as the instruments used. With respect to the latter, statistical information on their internal consistency and description has been included. The data analysis is also well developed and provides useful information for replicating the study.

  • Thank you

RESULTS

The results presented by the authors are closely related to the line of work proposed and its objectives.

  • Thank you

The section on Participants (together with Table 1) could be included within the Method since it performs a descriptive analysis of the sample.

  • We found examples in the journal that also put the description of the participants in the Results section.

The Tables are clear and allow understanding of the results.

  • Thank you

DISCUSSION

The authors discuss the results, assessing the verification of the hypotheses and comparing the information with other studies in an appropriate manner. Since a multi-author approach has been used in the theoretical framework (introduction), more use could be made of this valuable resource in the discussion as well, so it is encouraged to use more references than those previously cited in the Introduction in the Discussion if possible.

  • This was added where additional references were found.

The limitations are very well developed and allow us to analyze the work and the subject matter from a broader and more mature point of view. This subsection -just as a comment- could be called "Limitations and future lines of research".

  • Good suggestion. This was done.

CONCLUSIONS

The inclusion of this section is considered very opportune.

  • Thank you

REFERENCES

Adequate. They are in line with the subject matter.

  • Thank you

In summary, this is a very interesting work, with a good theoretical and empirical foundation.

  • Thank you